# Large-Scale Purification and Characterization of Recombinant Receptor-Binding Domain (RBD) of SARS-CoV-2 Spike Protein Expressed in Yeast

**DOI:** 10.3390/vaccines11101602

**Published:** 2023-10-16

**Authors:** Gaurav Nagar, Siddharth Jain, Meghraj Rajurkar, Rakesh Lothe, Harish Rao, Sourav Majumdar, Manish Gautam, Sergio A. Rodriguez-Aponte, Laura E. Crowell, J. Christopher Love, Prajakta Dandekar, Amita Puranik, Sunil Gairola, Umesh Shaligram, Ratnesh Jain

**Affiliations:** 1Serum Institute of India Pvt. Ltd., Hadapsar, Pune 411028, India; gaurav.nagar@seruminstitute.com (G.N.); sunil.gairola@seruminstitute.com (S.G.); 2Department of Biological Engineering, Koch Institute for Integrative Cancer Research, Massachusetts Institute of Technology, Cambridge, MA 02139, USA; sergrodz@mit.edu; 3Department of Chemical Engineering, Koch Institute for Integrative Cancer Research, Massachusetts Institute of Technology, Cambridge, MA 02139, USA; laura@sunflowertx.com (L.E.C.); clove@mit.edu (J.C.L.); 4Department of Pharmaceutical Sciences and Technology, Institute of Chemical Technology, Matunga, Mumbai 400019, India; pd.jain@ictmumbai.edu.in; 5Department of Biological Sciences and Biotechnology, Institute of Chemical Technology, Matunga, Mumbai 400019, India

**Keywords:** COVID-19, SARS-CoV-2, receptor-binding domain (RBD), vaccine, chromatography, characterization, angiotensin-converting enzyme 2 (ACE-2)

## Abstract

SARS-CoV-2 spike protein is an essential component of numerous protein-based vaccines for COVID-19. The receptor-binding domain of this spike protein is a promising antigen with ease of expression in microbial hosts and scalability at comparatively low production costs. This study describes the production, purification, and characterization of RBD of SARS-CoV-2 protein, which is currently in clinical trials, from a commercialization perspective. The protein was expressed in *Pichia pastoris* in a large-scale bioreactor of 1200 L capacity. Protein capture and purification are conducted through mixed-mode chromatography followed by hydrophobic interaction chromatography. This two-step purification process produced RBD with an overall productivity of ~21 mg/L at >99% purity. The protein’s primary, secondary, and tertiary structures were also verified using LCMS-based peptide mapping, circular dichroism, and fluorescence spectroscopy, respectively. The glycoprotein was further characterized for quality attributes such as glycosylation, molecular weight, purity, di-sulfide bonding, etc. Through structural analysis, it was confirmed that the product maintained a consistent quality across different batches during the large-scale production process. The binding capacity of RBD of spike protein was also assessed using human angiotensin-converting enzyme 2 receptor. A low binding constant range of KD values, ranging between 3.63 × 10^−8^ to 6.67 × 10^−8^, demonstrated a high affinity for the ACE2 receptor, revealing this protein as a promising candidate to prevent the entry of COVID-19 virus.

## 1. Introduction

The emergence of severe acute respiratory syndrome coronavirus 2 (SARS-CoV-2) represents the third highly pathogenic coronavirus to spread among the human population [1]. It belongs to the β-coronavirus genus—a member of the SARS-related coronavirus category [2]. SARS-CoV-2 has led to a disruptive global viral pandemic through the severe respiratory disease known as COVID-19 [2]. A coronavirus comprises four structural proteins viz. nucleocapsid (N), membrane (M), envelope (E), and spike (S) proteins. Spike protein is critical in viral attachment, fusion, and entry. Thus, it is an essential component of serological assays and a target component for developing entry inhibitors, antibodies, and vaccines [3]. The spike protein has two subunits, viz. S1 and S2. The receptor-binding domain (RBD) of the S1 subunit binds with the angiotensin-converting enzyme 2 (ACE2) receptor in the host cell to facilitate the viral entry into the host cell and S2 subunit, enabling the fusion of viral and host membrane [4]. Thus, RBD in SARS-CoV-2 S protein is an appealing target for spike protein-based vaccine designs [5]. RBD comprises 220 amino acids, two N-glycosylation sites (N331, N343), and nine cysteine residues [6,7,8,9]. While many researchers and some commercial manufacturers of RBD protein subunit vaccines have employed mammalian or insect cell line-based expression systems, the production cycles of these two vaccine platforms are relatively long and expensive [10]. Although mammalian and insect production platforms have shown effectiveness, it is challenging to implement them in existing large-scale microbial fermentation facilities located in low- and middle-income countries [11,12]. On the other hand, the *Pichia pastoris* expression system produces SARS-CoV-2 RBD-based recombinant protein in a cost-effective manner during large-scale manufacturing. This method does not usually require frozen storage and distribution of the resulting protein, which is safe and effective when combined with adjuvants [11].

Some of the reports indicate the suitability of *E. coli*, one of the most widespread bacterial hosts, for expressing heterologous proteins [10]. This approach has been used worldwide for cost-effective antigen-based serological testing. However, RBD of SARS CoV-2 expressed in *E. coli* lacks glycosylation and disulfide bonding, which is typically associated with significantly improved solubility and structural stability [10,13]. Furthermore, proteins derived from *E. coli* require refolding since they are expressed in inclusion bodies and are often contaminated with endotoxins, making their purification challenging [14]. Consequently, an alternative vaccine platform, such as a yeast expression system, could facilitate fast growth and rapid production at a relatively lower cost [15]. Yeast has a performance history of serving as a host organism for producing numerous pre-qualified and regulatory-approved recombinant subunit vaccines, such as influenza B, human papillomavirus, diphtheria, tetanus, hepatitis B, etc. [16]. Eukaryotic expression systems, such as yeast, offer numerous benefits over prokaryotes, which include better support towards post-translational modifications, protein folding, disulfide bridge formation, and secretory cleavage [15,16,17]. The robustness of the cells with low-cost production and full scalability further serves as icing on the cake. Therefore, an engineered yeast strain was employed for the improved secretion of the SARS-CoV-2 RBD, distinct from the circulating variants of Wuhan B.1.1.7, B.1.351, and Hu-1 virus strains [18].

*Pichia pastoris (Komagataella phaffii)*, a methylotrophic yeast, is regularly used to produce therapeutic glycoproteins at larger volumes due to its high-capacity eukaryotic secretory pathways [18]. Owing to the challenges posed by methanol in large-scale manufacturing facilities (such as flammability concerns and heat generation during fermentation), we worked on specifically tailored methanol-free yeast strain, reported previously [18]. 

To address the global COVID-19 pandemic, achieving the highest possible level of vaccination is essential. This scale ultimately implies the ability to produce billions of doses of these vaccine candidates. In this study, the RBD of SARS CoV-2 protein, which represents an antigen of interest, was expressed with SPYTAG peptide in an engineered *Pichia pastoris* for the cost-effective and rapid production of vaccine component at a 1200 L scale. A purification scheme comprising a mixed mode and hydrophobic interaction-based column chromatography process was developed and executed for four production batches. The purified RBD protein subunit as a potential vaccine candidate was characterized for its structural and functional properties by liquid chromatography hyphenated with high-resolution mass spectrometry LC-HRMS, CD spectroscopy, fluorescence spectroscopy, size exclusion chromatography (SEC), and bio-layer interferometry (BIL). This study demonstrated a high yield of ~21 mg/L for RBD, produced at 1200 L fermentation scale, in the engineered *Pichia pastoris* expression system. The purification process reported here yielded more than 99% purity in each batch with excellent similarity. This is the first report on the purification and characterization of RBD produced on a large scale (1200 L fermentation scale) in the *Pichia pastoris*-based expression system. 

The quality and quantity of the immune response can be efficiently improved by presenting the antigens to the immune system on virus-like particles (VLPs) [19]. The repetitive structures of VLPs facilitate their effective engagement with B cell receptors, while their size enables effective circulation to drain lymph nodes [20]. Various techniques are available to organize VLPs with the antigen(s) of interest, including genetic fusion, chemical derivatization, conjugation, or “plug-and-display” decoration. One method for conjugation with plug-and-display is forming a spontaneous iso-peptide bond between a peptide and its protein conjugate derived from a specific domain of a particular bacterial protein [20,21].

## 2. Materials and Methods

### 2.1. Materials

Capto-MMC and Phenyl sepharose 6FF were purchased from Cytiva, Uppsala, Sweden. Biotinylated ACE2 receptor protein and streptavidin-coated biosensors were procured from ACRO Biosystems and Sartorius (previously known as Forte Bio, Newark, DE, USA), respectively. Urea, Ethylene-diamine-tetra-acetic acid, i.e., EDTA, sodium citrate, sodium chloride, sodium dihydrogen phosphate, disodium hydrogen phosphate, citric acid, and other chemicals for analytical and purification purposes were purchased from Sigma-Aldrich (St. Louis, MO, USA). 1200 L customized bioreactor was purchased from Bioengineering, Wald, Switzerland, the continuous centrifugation (CSE 100) system was from GEA Westfalia (Oelde, Germany), while the AKTA Process 10 mm PP Gradient chromatographic purification system was from Cytiva (Uppsala, Sweden). BMGY media was prepared in-house by using BD-yeast extract and BD-peptone, glycerol from Avantor, Biotin from Sigma-Aldrich, yeast nitrogen base from Himdeia, and Potassium phosphate from ThermoFisher and Kronox life sciences, respectively. The ultrafiltration system was procured from Pall (New York, NY, USA), while Pellicon-2 (5 KDa) ultrafiltration cassettes were purchased from Millipore (Burlington, MA, USA). Liquid chromatographic system (1290 series) hyphenated with mass spectrometry (6540 UHD Accurate mass) and all chromatographic columns, such as C4 and C18 RPLC columns, were from Agilent Technologies, Santa Clara, CA, USA. Mass spectrometry grade enzymes such as PNGase F and Glu-C were from Thermo Scientific, while trypsin was from Sigma-Aldrich. 

### 2.2. Methods

#### 2.2.1. Expression and Production of RBD in *Pichia Pastoris*

All strains and vectors used for the production of the protein used in this study were from the collection of the Alternative Host Research Consortium at MIT (AltHost). The strain used for the expression and production of RBD was derived from the wildtype *K. phaffii* (USDA NRRL Y-11430) [22]. Strain Information RNA sequencing data are available in the NCBI Gene Expression Omnibus, accession GSE183408. The RBD containing gene transformation and modification to reduce expression and production challenges have already been reported before, and the same construct vector (AltHost L vector, D-320) with strain was used in this study [18,23]. The gene that carries the RBD was optimized for codons, synthesized by Integrated DNA Technologies, and then inserted into a custom vector [23]. The Appendix A contain information on vector construction and strain details (Appendix A). Methanol-inducible promoter P_AOX1_ was used to express the recombinant gene (RBD) with methanol feed. However, due to heat generation and flammability of methanol, it would be challenging to implement at large-scale manufacturing. An additional copy of *mit1* using the P_CAT1_ promoter and *TEF1* terminator was inserted at an intergenic site near GQ67_02967 to yield a methanol-free engineered strain of *Pichia pastoris* (*Komagataella phaffii* (AltHost strain S-380) carrying a gene for the receptor-binding domain (RBD) of SARS-CoV2 fused to a 13 amino acid polypeptide derived from *Streptococcus pyogenes* known as SPYTAG [18,23,24]. The cell bank was converted into master working cell banks (MWCBs), which were assessed further to meet specifications for using them during product manufacturing. RBD Spy tag protein was thus produced using these WCBs by fermentation at a 1200L scale at Serum Institute of India Pvt. Ltd. Further evaluation was conducted to assess its structural and functional properties. Buffered Glycerol Complex Media (i.e., BMGY media: 1% yeast extract, 2% bactopeptone, 1.34% YNB, 0.02% biotin, 0.1 M potassium phosphate, pH 6.0, and 1% glycerol) was used as the primary fermentation media with Glycerol as a carbon source and Sorbitol as a protein expression inducer (media details of fermentation are given in Appendix A).

(i) Initially, 1 mL MWCB was inoculated in a 100 mL shake flask containing BMGY medium to achieve an optical density of approximately 10 absorbance units (AU) at 600 nm (OD600); (ii) Once the desired OD600 was achieved, the culture in the shake flask was transferred to a new 4 L shake flask to double the OD600 value from the previous step; (iii) A 170 L seed fermenter was filled with 90 L media and inoculated with culture from a 4 L shake flask to achieve the final OD600 of approximately 30 AU. At this stage, agitation was provided in the cascade mode to maintain DO at 25%. The temperature was maintained at 25 °C, and the pH was maintained at 6.5; (iv) A 1200-L fermenter containing approximately 1000 L of the fermentation medium was inoculated with the seed culture. This was followed by agitation in cascade mode to ensure optimum conditions for fermentation. DO was maintained at 25%, with 50% glycerol substrate feed. The temperature was maintained at 25 °C, and the pH was maintained at 6.5. After achieving OD600 ≥ 40 AU, protein expression was induced by the addition of 50% sorbitol stock solution to achieve a final concentration of 1% sorbitol in the fermenter culture after approximately 24 h EFT (elapsed fermentation time). Subsequently, the culture was harvested ~24 h post-induction by reducing the process temperature to 8–12 °C (refer to Appendix A for detailed fermentation parameters). The harvested biomass (1100–1200 L) exhibited OD600 in the range of 50–60AU and pack cell weight in the range of 0.12–0.15 g/g.

#### 2.2.2. Harvest and Purification of RBD

6 M urea and 0.5 M EDTA were added to the culture harvest to achieve the final concentration of 1 M urea and 10 mM EDTA to control the proteolytic activity, preventing aggregation and increasing the protein solubility and stability in the culture supernatant. Approximately 220–240 L of 6 M urea were added to an estimated harvest volume of 1100–1200 L. This mixture was continuously centrifuged at 6750 to 8000 RPM (x1.81 for conversion into RCF) with a vessel radius of 22.3 cm to separate cell debris from the supernatant that comprises the desired protein (as secretory protein expression system employed in this process) [18]. The partial ejection time for centrifugation was 400 s, with back pressure maintained at 6.5 bar. Approximately 800 L of culture supernatant was collected after centrifugation. 1 M citric acid was added to the centrifuged cell-free supernatant to attain pH 5.0, and the mixture was further clarified by sterile filtration with Opticap XL-5 Durapore filter by Merck Millipore. This step assisted in reducing the bioburden before proceeding to downstream processing. 

Capto MMC (Cytiva, Uppsala, Sweden) mixed-mode cation exchange chromatography with approximately 43 L column volume was used as an initial purification step to capture the desired protein. Phenyl Sepharose 6 FF resin (Cytiva, Uppsala, Sweden) hydrophobic interaction chromatography with approximately 52 L column volume was used for the polishing step to get a purity of more than 95%. Column packing was carried out in-house using a Chromatography column Packing station by GE Healthcare, with a flow rate of 10–50 L/min. Column packing efficiency was evaluated using Asymmetry and HETP as criteria per the manufacturer’s instructions. These steps assisted in achieving maximum recovery of RBD SARS-CoV2 protein with SpyTag from the culture supernatant and improved target protein purity. Operating steps with corresponding buffer, volume, and flow details of both chromatography are provided in Table 1 and Table 2.

The peak corresponding to RBD (50–55 L) was collected and concentrated to 2–4 L. It was then processed via diafiltration and concentration to achieve a final concentration of 10–15 mg/mL, using 5 KDa PES (Polyethersulphone) cassettes in tangential flow filtration (TFF), as measured at A280. The buffer used for concentration–diafiltration contained 0.1 M sodium chloride, 5 mM EDTA, and 20 mM sodium phosphate buffer to achieve a pH of 7.4. This process resulted in the final drug substance at the concentration of 10–15 mg/mL and from 1.68 L to 2 L. The total batch process time at 1200 L fermentation scale, since inoculation of shake flask to the concentration and diafiltration steps (drug substance), was ~6 days. It was then validated to ensure it met pre-defined quality specifications, including a purity level of over 99%, as confirmed by SDS-PAGE and SE-HPLC.

Yield was calculated by dividing the total protein (mg) of the final drug substance by fermenter harvest volume (L). In the current study, total protein and harvest volume ranged from 23.52 to 27.20 g and 1192 to 1200 L, respectively. Hence, the yield ranged from 19.73 to 22.67 mg/L, with an average of 21.08 mg/L.

#### 2.2.3. SDS-PAGE and Western Blot Analysis

The purified final drug substance was analyzed by using SDS-PAGE (NuPAGE™ 4—12%, Bis-Tris, 1.0 mm) mini protein gels under the reducing conditions (5% beta-mercaptoethanol, i.e., BME *v*/*v*) followed by Coomassie blue R 250 staining.

For deglycosylation of RBD protein, 15 µg of intact RBD protein sample at the concentration of 2 mg/mL was added with 6 µL of GlycoWorks buffer (Waters). The mixture was denatured by heating it for 3 min on a heating block until the temperature reached at least 90 °C. Thereafter, 1 mL tube was removed from the heating block and allowed to cool for 3 min. Further, 1.2 µL of PNGase-F enzyme (NEB) was added to the reaction mixture and incubated at 50 °C for 5 min, further cooled at room temperature (~25 °C) for 3 min before being analyzed by SDS-PAGE.

Polyclonal primary antibodies of rabbit origin (from Sino Biological Inc., Beijing, China) against SARS CoV2/2019nCOV Spike/RBD were used for identification by Western blotting. Primary antibodies were diluted to 1:10,000 concentration. Anti-rabbit alkaline phosphatase-conjugated antibodies (Invitrogen™, ThermoFisher Scientific, Waltham, MA, USA, part no. 46-7007) were used as secondary antibodies. The colorimetric detection was carried out using Chromogen (BCIP/NBT) Invitrogen™ (ThermoFisher Scientific, Waltham, MA, USA, Part no: SO -W001). 

#### 2.2.4. Intact Protein Analysis and Peptide Mapping by Using LC-MS

To analyze intact proteins, a sample containing 25 μg of protein was treated with PNGase-F enzyme at a 1:20 enzyme-to-protein ratio for 14 to 24 h at 37 °C to remove glycans. The glycans that were released were separated from the protein using ethanol precipitation. The remaining protein pellet was dissolved in HPLC-grade water and incubated with 50 mM DTT for 30 min at room temperature (25 ± 2 °C) before analysis. 15 μL of this protein solution (~30 μg) was injected over a C4 RPLC column (AdvanceBio RP-Mab by Agilent Technologies—USHFP01121) for intact level protein analysis. Mobile phase A consisted of 0.1% formic acid in 5% Acetonitrile and 95% water solution, while mobile phase B was composed of 0.1% formic acid in a solution of 90% acetonitrile and 10% water. Chromatographic separation was carried out using an acetonitrile gradient (5 to 80%), and the eluate was infused online to the ESI-Q-TOF mass spectrometry set-up. Mass spectrometric data acquisition was carried out in positive mode for a mass range of 300–6000 *m*/*z* with a scan rate of 1 spectra/s. Drying and sheath gas flow rates were 11 L/min, the nebulizing gas pressure was 50 psig, and the sheath gas temperature was 400 °C. Deconvoluted spectra were processed for the evaluation of molecular mass by using Promass™ software (Novatia LL). 

Peptide mapping was carried out by taking around 25 μg of protein sample and treating it with PNGase F enzyme for 14 to 24 h at 37 °C for deglycosylation. Released glycans were separated through ethanol precipitation followed by reconstitution of protein pellet in 8 M Urea/50 mM ammonium bicarbonate (NH_4_CO_3_). Proteins were reduced by adding 100 mM DTT/50 mM NH_4_CO_3_ for 30 min at 56 °C. The sample was further incubated for 15 min in the dark after adding 200 mM iodoacetamide/50 mM NH_4_CO_3_. Additionally, this protein sample was divided into two different vials, and proteolysis was carried out simultaneously by adding trypsin to one tube and Glu-C enzyme to another, in a ratio of 1:20 (Enzyme: substrate). Proteolysis was carried out for 16 h at 37 °C, and the reaction was then quenched by acidifying the solution with the addition of TFA. The resultant peptide mixture was separated using C18 RP-HPLC with eluate infused online with an ESI-QTOF mass spectrometer. Chromatographic analysis was carried out at a flow rate of 0.3 mL/min with 5% of mobile phase B from 0 min to 48 min, 5% to 100% B from 48 min to 52 min with 100% B till 56 min for washing followed by re-equilibration for 4 min at 5% mobile phase B. Mass spectrometric data acquisition was carried out in positive ionization mode with drying gas flow rate of 11 L/min while sheath gas flow rate of 10 L/min. The nebulizing gas pressure was 25 psig with 290 °C temperature, while the sheath gas temperature was 295 °C. Acquired data were processed for the evaluation of sequence coverage by using the Morpheus software tool (V—272; Open Source https://cwenger.github.io/Morpheus/ accessed on 1 March 2021). 

#### 2.2.5. Glycosylation Analysis

The RBD sample was diluted in 8 M Urea/50 mM NH_4_CO_3_, followed by a reduction of protein by adding 100mM DTT in 50 mM NH_4_CO_3_ for 30 min at 56 °C temperature. Further, 200 mM Iodoacetamide /50 mM NH_4_CO_3_ was added to the sample solution, and samples were incubated in the dark at room temperature for 15 min. Protease Trypsin was added in the ratio of 1:20 (Enzyme: Substrate), and proteolysis was carried out for 16 h at 37 °C. After 16 h incubation, samples were acidified by adding TFA. Glycopeptides were concentrated using speed-vac and dissolved in 0.1% formic acid (FA). MS and MS/MS data were acquired with a mass range of 300–6000 *m*/*z*, and the data were extracted using Maxquant software. The peptide mass information was compared with the theoretical MS profile of trypsin-digested protein using GlycoMod tools. Mass searches were carried out at a precursor mass threshold of <0.005 Dalton using ‘Carbamidomethylation’ of cysteine as a fixed modification. 

#### 2.2.6. Size Exclusion Chromatography Analysis

Size exclusion chromatography (SEC) analysis of RBD was carried out on a 1290 Agilent liquid chromatographic system with AdvanceBio SEC 300 °A, 2.7 µm, and 4.6 × 300 mm column (PL1580-5301—Agilent Technologies, USA). The analysis was carried out in isocratic mode using 150 mM Sodium Phosphate, pH 7.0, at a constant flow rate of 0.25 mL/min. A total of 10 micrograms of protein sample were injected into the column, and the analysis was performed at 280 nm using a UV detector. 

#### 2.2.7. Di-Sulphide Bond Analysis

The protein sample was deglycosylated using PNGase F (enzyme-to-protein ratio of 1:20) and digested with Trypsin protease (1:20 enzyme-to-protein ratio) in non-reducing conditions to preserve the Di-Sulfide bonds. Obtained peptides were analyzed using MaxQuant. The obtained mass values were compared with the theoretical peptide mass obtained from the protein in non-reducing conditions, assuming the presence of Di-Sulfide bonds at the expected location. A stringent mass match threshold of <0.05 Dalton was utilized for the searches.

#### 2.2.8. Far-UV Circular Dichroism Spectroscopy

The secondary structure of RBD was evaluated by the far-UV circular dichroism (CD) spectroscopy. Far-UV CD spectra (190 nm to 250 nm) of the samples were collected using a Jasco J-1500 CD spectrophotometer (Tokyo, Japan) in blank subtracted mode. 50 mM phosphate buffer was used as a blank. Samples were diluted at 100 μg/mL concentration in 50 mM phosphate buffer, pH 7.4. An average of five spectral scans were accumulated. Ellipticity values were corrected for the protein concentration and the cuvette’s path length. Dichroprot software was used to analyze the CD data, and data points were fit with a constrained least square method to obtain the value of alpha-helical, beta-pleated sheet, and other (turn and coil) secondary structure components.

#### 2.2.9. Fluorescence Spectroscopy

Samples were diluted using 50 mM phosphate buffer in 500, 250, and 125 μg/mL concentrations. A 10 µM concentration of tryptophan amino acid was prepared in 50 mM phosphate buffer as a reference standard for comparative study. Protein samples were excited at 295 nm to specifically excite tryptophan and emission spectra were collected using a Horiba Scientific Floromax-AC spectrophotometer in the 300–450 nm range. The fluorescence intensity of samples and Tryptophan was normalized, and the spectral patterns were evaluated. 

#### 2.2.10. RBD Spy Tag Binding Kinetics with ACE2 Receptor Protein Using Bio Layer Interferometry (BLI)

Bio-layer interferometry was used to measure the affinity binding constants of the RBD spy tag. All assay conditions were prepared in a Greiner 96-well plate (#655209) in a volume of 250 µL using Kinetic assay buffer (1X PBS containing 0.05% Tween-20 including 0.5%BSA, pH 7.2). Biotinylated ACE-2 human protein (AC2-H82E6, ACRO Biosystems) with a C-terminal AviTag was diluted into assay buffer at 10 µg/mL and immobilized onto streptavidin-coated biosensors (#18-5019, Forte Bio) to a minimum response value of 1 nm on the Octet Red96 System (Forte Bio). A baseline response was established in the assay buffer before each association. The RBD spy tag was diluted into assay buffer at a 10–300 nM grade concentration. The RBD spy tag was allowed to associate for 200 s, followed by dissociation for 600 s in the same baseline wells. The assay included one biosensor with only assay buffer, which was used as the background normalization control. Using the Forte Bio Data Analysis suite, the data were normalized to the association curves following background normalization and Savitzky–Golay filtering. Curve fitting was applied using a 1:1 interaction model with the global fitting of the sensor data, and a steady state analysis was used to determine the association rate constant (kon), dissociation rate constant (koff), and equilibrium dissociation constant (KD).

## 3. Results

### 3.1. Purification of RBD Protein

The manufacturing scheme at the 1200 L fermentation scale is shown in Figure 1. Purified bulk RBD protein samples using multi-dimensional chromatographic stages (Appendix A) from four production batches were analyzed by SDS-PAGE (Figure 2A,B) and Western blot (Figure 3). The SDS-PAGE analysis (4–12% Bis-Tris) of 5 µg purified RBD was found to result in a single distinct protein band, with an apparent molecular weight of ~28.5 kDa, under reduced conditions (glycosylated) (Figure 2A and Figure 3), indicating the high purity of the RBD SARS-CoV2 antigen (Figure 2B). Clear distinguishable bands of glycosylated and deglycosylated forms of RBD were observed at ~28.5 kDa and ~24.5 kDa, respectively, with remarkable differences of ~4 kDa in Figure 2A,B. Further, Western blot analysis of a clear, distinct band confirmed the identity against RBD-specific antibody (Figure 3).

### 3.2. Intact Mass Analysis by LC-HRMS

Purified RBD of SARS-CoV2 protein bulk from four different batches manufactured at ~1200 L scale was evaluated for molecular mass as 24,658.7 Da, 24,659.4 Da, 24,659.6 Da, and 24,660.2 Da under deglycosylated and reduced conditions using LC–MS (Figure 4A–D).

### 3.3. LC–MS-Based Peptide Mapping

Primary structure (i.e., the amino acid sequence of the RBD protein) was evaluated using LC–MS for tryptic digests of protein samples from four batches (Figure 5). All samples demonstrated 100% amino acid sequence coverage with the delta mass (i.e., the difference between the theoretical and observed peptide mass) within 20 ppm. Thus, the amino acid sequence coverage and base pair ion chromatogram indicate significant similarity among all four batches (Appendix A). 

### 3.4. Size Exclusion Chromatography Analysis

Size exclusion chromatography (SEC) was carried out for evaluation of the presence of any high-molecular-weight (HMW) or low-molecular-weight (LMW) species apart from the target monomer of the RBD molecule in the given samples. All four batches demonstrated similar features with no significant peaks observed other than a monomeric peak (Figure 6). These data suggest minimal high- or low-molecular-weight species in the purified protein.

### 3.5. Secondary Structure Analysis by Circular Dichroism Spectroscopy

CD spectrum of the purified protein samples showed a dominant negative peak at ~209 ± 1nm. The analysis of secondary structure through Dichroprot software suggested that purified RBD of four different batches consists of the alpha helix (6–7%), beta sheet (25–26%), beta turn (48–49%), and random coil (18–19%). A summary of batch-specific secondary structural component contribution is provided in Table 3 and Figure 7, demonstrating significant similarity among all four batches of RBD (Figure 7).

### 3.6. Tertiary Structure Analysis by Circular Dichroism Spectroscopy

The tertiary structure of the protein samples was evaluated by referring to the fluorescence emission pattern (emission maxima) of intrinsic tryptophan (present in the protein). The signal was compared with the emission pattern of tryptophan in aqueous solution. Exciting the sample obtained fluorescence signals at 295 nm, which Tryptophan specifically absorbs; compared to free Tryptophan, a clear blue-shifted emission maximum (λem, max) from the protein sample suggested that Tryptophan is buried in the hydrophobic pocket (Figure 8). This demonstrated that the protein was present in a folded state. The λem, the max value of different samples, was found to be 336 ± 1 nm. The λem, max value, and spectral pattern of each sample were similar, indicating the similarity of the tertiary structure among all four batches analyzed.

### 3.7. Di-Sulfide Bond Analysis Using LC-MS

Di-sulfide bonds have a critical role in maintaining the higher-order structure of the protein. As expected, LC-HRMS was used to analyze and confirm the presence of four disulfide bonds in the protein. Protein samples were digested with Trypsin in both non-reducing and reducing conditions to facilitate the detection of disulfide-linked peptides present in the Trypsin digest. Table 4 provides a comparative summary of theoretical vs. observed masses of disulfide-like peptides. These results demonstrated that the proteins from all four batches possessed the expected numbers of disulfide bonds, thus likely retaining precise higher-order structure (Appendix A).

### 3.8. Glycosylation Analysis Using LC-MS

N-linked Glycopeptide analysis was then carried out for protein samples from all four batches. The obtained mass values of the peptides were fed to the Expasy Glycomod glycopeptide database and mass searches were carried out at a stringent mass match threshold of <0.005 Dalton. Among several peptides feasible upon protein digestion, the mass of peptide ITNLCPFGEVFNATR (amino acid 1 to 15) that harbors glycan conjugation sites at N-12 was matched. The identified Glycans in samples are listed in Appendix A.

### 3.9. Binding Kinetics Evaluation by Using Bio-Layer Interferometry (BLI)

Four different batches (12680T002, 12680T003, 12680T004, 12680T005) of RBD spy tag binding kinetics interactions with human ACE2 receptor were purified using BLI under identical conditions and demonstrated a similar association rate constant (kon), dissociation rate constant (koff), and equilibrium dissociation constant (KD) (Figure 9, Table 5). The Chi q2 and R2 values were <1 and >0.9, respectively, in each batch, which further attested to the validity of the binding kinetic data. No significant log differences were found among the four batches in the KD, Kon, and Koff values. This observation confirmed the similar binding affinity with no structural variability at the epitope sites of RBD spy tag protein in all batches.

## 4. Discussion

Numerous vaccine candidates based on the RBD protein antigen have been developed recently due to the unprecedented COVID-19 global pandemic. Many RBD antigen-candidate vaccines were expressed in mammalian or bacterial cells [14]. However, RBD production in a mammalian expression system has limitations of higher operating costs compared to microbial production [25]. Thus, a yeast expression system was evaluated and scaled up from a 20 to 1200 L scale for the production of RBD at a target cost of goods manufactured (COGSm). RBD produced from mammalian (HEK-293T) and yeast (*P. pastoris*) are broadly similar in molecular weight after treatment with PNGaseF, and both the polypeptides have similar structures [26]. The amino acid sequence of the RBD Spike protein is considered identical to the RBD region of reference sequence YP_009724390.1 publicly available in the gene bank (accessed on 5 May 2022 https://www.ncbi.nlm.nih.gov/ipg/YP_009724390.1). In this study, the yield of RBD was ~21 mg/L at a 1200 L fermenter scale. Thus, the study confirmed the proposed process’s scalability from 20 L to 1200 L fermentation scale. Evaluation of structural and functional properties for four manufacturing batches resulted in ≥99% and demonstrated product consistency with remarkable batch-to-batch similarity at 1200 L scale. Various analytical approaches, such as SE-HPLC, SDS-PAGE, and Western blot analysis, demonstrated that the purity and identity of four batches of RBD were identical (Summary of batch release data Appendix A). Remarkably, the mass difference between glycosylated and deglycosylated forms of RBD in SDS-PAGE analysis denoted the presence of N-linked glycan in the molecular organization. This glycan moiety may have played a vital role in the native conformation, stability, and functional activity of the molecule [27]. Due to hyper-glycosylation, the first glycosylation site (N331) was omitted in the construct. N343 alone maintained the required properties of RBD protein [28]. The theoretical mass of the protein estimated based on its amino acid sequence was 24,784 Da, while the observed mass in LC–MS (Deglycosylated) was found to be 24,658.7, 24,659.4, 24,659.6, and 24,660.2 Da. The difference of approximately 125 Da between observed and theoretical mass was attributed to the ‘Lysine-loss’ at the C-terminus of the protein. The determined mass was similar for samples obtained from different batches, thus demonstrating appropriate protein translation and batch-to-batch consistency at a 1200 L scale. Further SEC analysis confirmed the absence of HMW and LMW impurities profiles closely resembled among the four batches.

These results demonstrated that the proteins from all four batches possessed the expected numbers of disulfide bonds, thus likely retaining a precise higher-order structure. N-linked Glycans of samples were analyzed using Mass spectrometry and chromatography-based methods. Asparagine amino acid at the 12th position was found to be glycosylated, and several simple and complex glycans were identified to be linked with it. The glycosylation site was the same in all the samples, and the type of glycans identified from different batches was also broadly similar. Hence, it was concluded that the glycosylation profile of protein obtained from different samples is likely to be similar. Further, glycosylation probably prevents the aggregation of proteins under native conditions in the extracellular matrix [29,30].

The secondary structure of four batches of RBD was identical, suggesting that beta-sheets were one of the closely resembling dominating secondary structure components of four batches of RBD protein at the far UV region. This demonstrates significant similarity among all four batches of RBD. Likewise, tertiary structure evolution also showed that the protein was present in a folded state.

Moreover, the binding profile demonstrated rapid association concerning early saturation, confirming the strong affinity of the RBD spy tag with ACE2 proteins, while instantaneous dissociation established irreversible binding. The binding of RBD protein with ACE2 receptors protein was used to understand the efficacy of RBD proteins since the mechanism of RBD proteins mediates the binding of the virus to the host cell, considered a crucial stage of entry in target cells. A high binding affinity, with KD values ranging between 3.63 × 10^−8^ M to 6.67 × 10^−8^ M, of all four batches of purified RBD could prevent binding of viral entry with the ACE2 receptor [31,32].

Studies conducted in macaques and mice have demonstrated the effectiveness of the produced RBD. The RBD SpyTag was conjugated with HBsAg SpyCatcher nanoparticles and formulated with alum during evaluation in cynomolgus macaques. The vaccine elicited a high titer of neutralizing antibodies (>10^4^). After being challenged with SARS-CoV-2, the vaccine was found to provide protection against viral loads in both the upper and lower respiratory tract. The SARS-CoV-2 receptor-binding domain displayed on HBsAg virus-like particles elicits protective immunity in macaques [11].

In another study, RBD HBsAg was proven as an immunogenic antigen. When adjuvanted with either alum or SWE, it offered protection to mice that were challenged with the alfa or beta variant. The protection profiles due to RBD HBsAg vaccines were comparable with mRNA Pfizer vaccination [33].

RBD-based COVID-19 vaccine utilizing VLP technology, developed by Serum Institute of India Pvt. Ltd. and SpyBiotech, has been registered in phase 1/2 clinical trials in Australia. (ACTRN12620000817943, ACTRN12620001308987) [34].

## 5. Conclusions

The receptor-binding domain (RBD) of SARS-CoV-2 spike protein was, successfully produced at a large scale in a *Pichia pastoris*-based expression system with a high yield of ~21 mg/L. The scheme of the purification process offered more than 99% purity in each batch with excellent batch-to-batch similarity. A comparability thorough characterization and testing for all representative batches confirmed that protein quality aligned with the expected parameters maintained structural integrity and demonstrated consistent binding affinity. Thus, a yeast-based expression system offers a highly effective and affordable option for producing RBD on a large scale.

## Figures and Tables

**Figure 1 vaccines-11-01602-f001:**
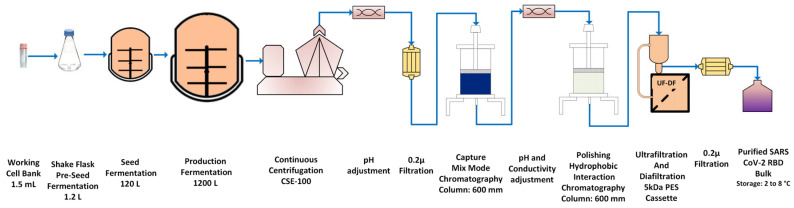
Schematic representation of manufacturing process flow of RBD.

**Figure 2 vaccines-11-01602-f002:**
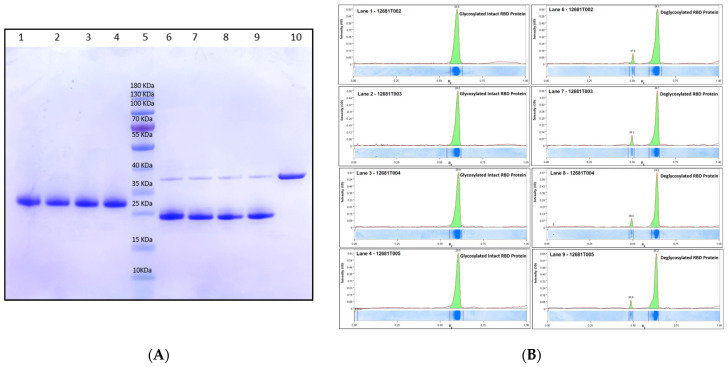
(**A**): Purity determination of four consistency production batches of RBD SARS-CoV2 antigen fused with SPYTAG by 4–12% SDS-PAGE under reduced conditions with Coomassie brilliant blue (CBB) staining of all four batches in glycosylated and deglycosylated form. Lane 1: Glycosylated Batch 12681T002, Lane 2: Glycosylated Batch 12681T003, Lane 3: Glycosylated Batch 12681T004, Lane 4: Glycosylated Batch 12681T005, Lane 5: Pre-stained protein standard marker, Lane 6: Deglycosylated Batch 12681T002, Lane 7: Deglycosylated Batch 12681T003, Lane 8: Deglycosylated Batch 12681T004, Lane 9: Deglycosylated Batch 12681T005. Lane 10: PNGase F enzyme; (**B**): Densitometric profile of Lane 1: Glycosylated Batch 12681T002, Lane 2: Glycosylated Batch 12681T003, Lane 3: Glycosylated Batch 12681T004, Lane 4: Glycosylated Batch 12681T005, Lane 6: Deglycosylated Batch 12681T002, Lane 7: Deglycosylated Batch 12681T003, Lane 8: Deglycosylated Batch 12681T004, Lane 9: Deglycosylated Batch 12681T005.

**Figure 3 vaccines-11-01602-f003:**
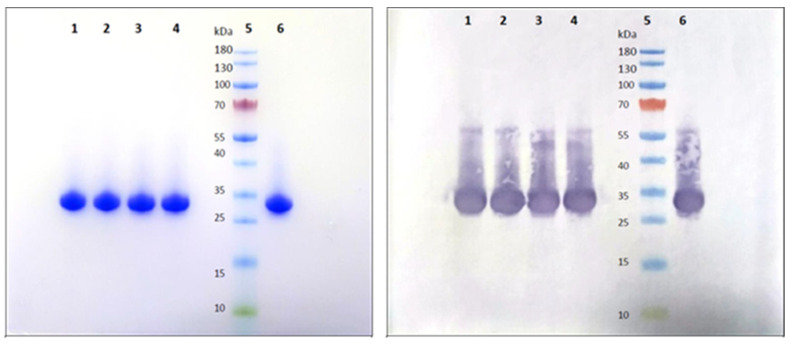
Determination of the identity of four consistency production batches of RBD SARS-CoV2 antigen fused with SPYTAG by Western blotting. Lane 1: Batch 12681T002, Lane 2: Batch 12681T003, Lane 3: Batch 12681T004, Lane 4: Batch 12681T005, Lane 5: Pre-stained protein standard marker and Lane 6: In-house standard RBDSARS-CoV2 antigen with Spy tag protein.

**Figure 4 vaccines-11-01602-f004:**
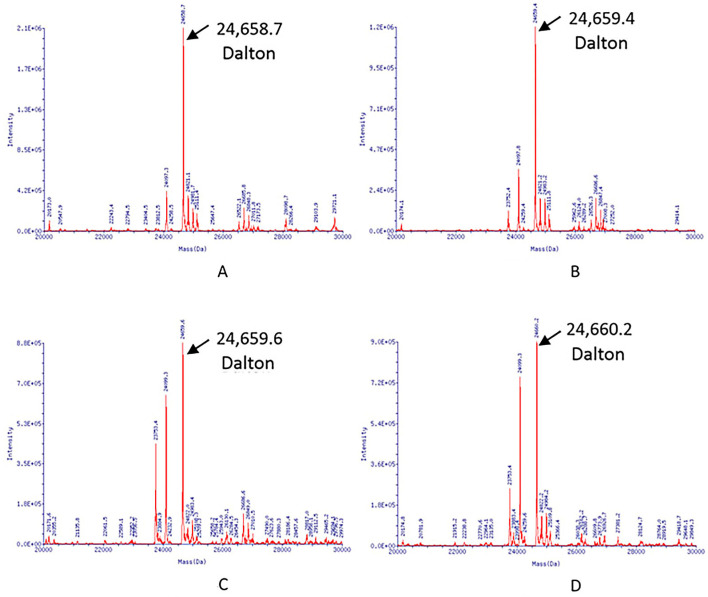
LC–MS spectrum of RBD of SARS-CoV2 protein (**A**) Batch 12681T002, (**B**) Batch 12681T003, (**C**) Batch 12681T004, (**D**) Batch 12681T005. The spectrum shows the signal at *m/z* of the spectra was deconvoluted using Promass software, and the molecular mass of the primary peak was determined to be 24,659 ± 1 Dalton, corresponding to a single species with a molecular weight of 24.6 kDa.

**Figure 5 vaccines-11-01602-f005:**
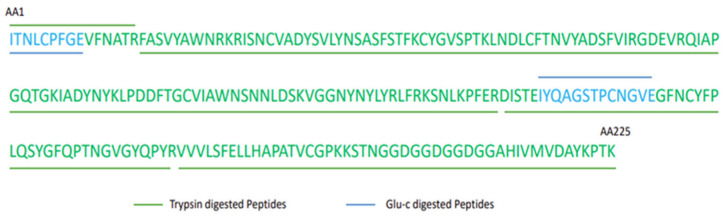
Amino acid coverage of purified RBD proteins of four consistency batches by peptide mapping.

**Figure 6 vaccines-11-01602-f006:**
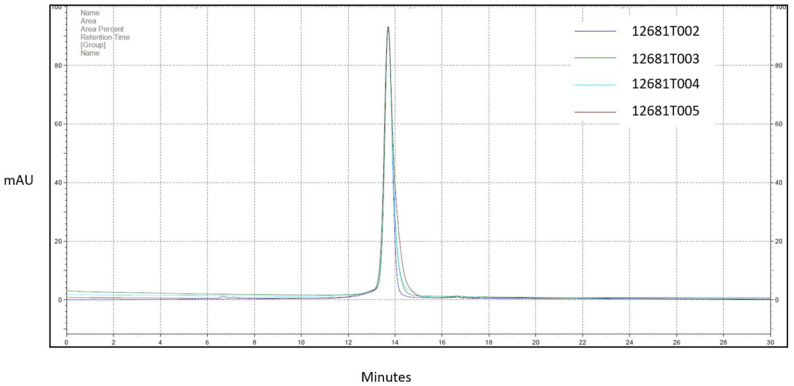
Summary of SEC-HPLC carried out for RBD samples from four production batches.

**Figure 7 vaccines-11-01602-f007:**
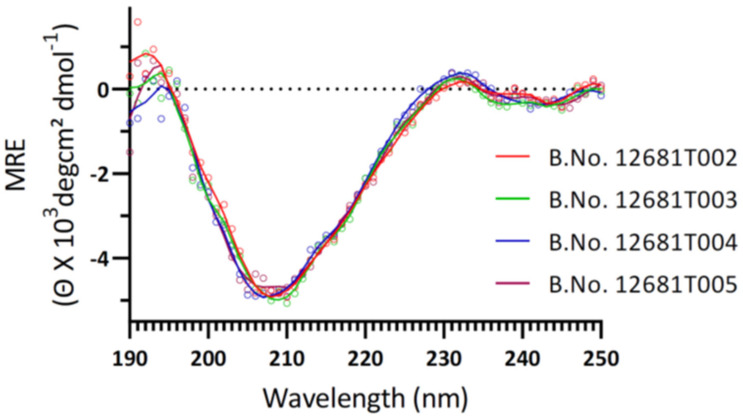
Determination of the secondary structure of purified RBD proteins of four consistency batches by circular dichroism (CD) analysis.

**Figure 8 vaccines-11-01602-f008:**
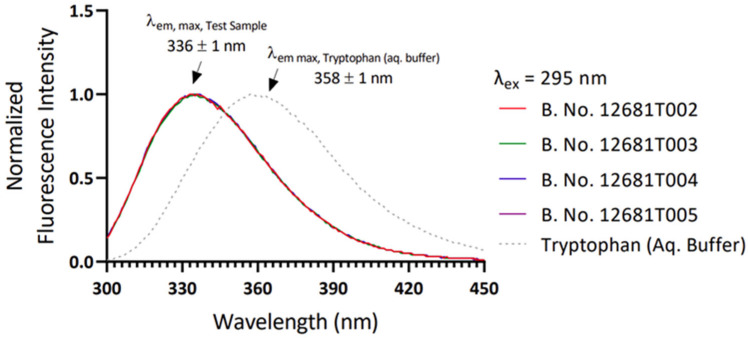
Comparison of fluorescence emission profile of four different batches of RBD.

**Figure 9 vaccines-11-01602-f009:**
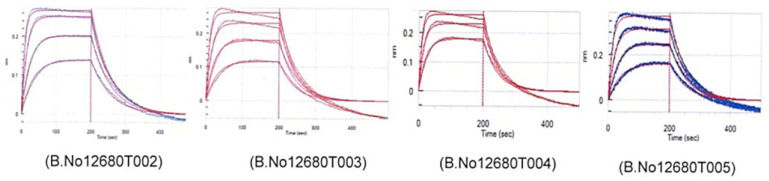
Binding profile of RBD spy tag batches (B.No 12680T002, 12680T003, 12680T004, 12680T005) with human ACE2 receptor protein on OCTET RED 96 platform at a concentration of analyte ranging from 28.4 nM to 227.3 nM. The red line is a global fit using the instrument line function.

**Table 1 vaccines-11-01602-t001:** Operating steps with description for mixed-mode cation exchange chromatography (Capto MMC).

Step	Flow Rate (cm/h)	Column Volume (L)	Description/Composition
Equilibration	226	05	20 mM Citrate buffer (pH 5.0)
Load	226	~25	Centrifuged and clarified fermentation supernatant adjusted to pH 5 ± 0.3.
Wash-1	226	05	20 mM Sodium Phosphate buffer pH 6.5 + 1.5 % Isopropanol (IPA) and 0.05% Triton-X-100—to strip of the endotoxins
Wash-2	226	05	Phosphate buffer (pH 7.0)—to strip off the traces of IPA and Triton-X from column
Elution	226	05	20 mM Sodium phosphate buffer (pH 8.0) + 0.5 M Sodium Chloride + 5 mM EDTA (Peak collected from the start in absorbance till fall)

**Table 2 vaccines-11-01602-t002:** Operating steps with description for hydrophobic interaction chromatography (Phenyl-Sepharose 6FF).

Step	Flow Rate (cm/h)	Column Volume (L)	Description/Composition
Equilibration	120	05	20 mM Sodium phosphate buffer (pH 6.5) + 0.5 M Sodium Chloride
Load	120	~4	Capture eluate pH adjusted to 6.2–6.8 and 50–70 mS/cm Conductivity.
Wash	120	05	20 mM Sodium phosphate buffer (pH 6.5) + 0.5 M Sodium Chloride
Elution	120	05	20 mM Sodium phosphate buffer (pH 7.4) + 5 mM EDTA—decreasing salt content to ~0.1 M with the simultaneous increase in pH. (Peak collected from the start in absorbance till fall)

**Table 3 vaccines-11-01602-t003:** Summary of the estimated structural component of four different batches of RBD by Circular dichroism spectroscopy.

Batch No.	Alpha-Helix	Beta Sheet	Beta Turn	Random Coil
12680T002	7%	25%	48%	18%
12680T003	8%	26%	48%	19%
12680T004	6%	26%	49%	19%
12680T005	7%	26%	49%	19%

**Table 4 vaccines-11-01602-t004:** Comparative summary of theoretical vs. observed masses of di-Sulfide-like peptides of purified RBD proteins.

Di-Sulfide Linked Peptides	Di-Sulfide Bond Position	Theoretical Mass (Dalton)	Observed Mass (Dalton)			
			B. No. 12681T002	B. No. 12681T003	B. No. 12681T004	B. No. 12681T005
ITNLCPFGEVFNATR + ISNCVADYSVLYNSASFSTFK	C5-C30	3993.92	3993.89	-	3993.91	-
ITNLCPFGEVFNATR + KRISNCVADYSVLYNSASFSTFK	C5-C30	4278.06	-	4278.10	-	4278.10
CYGVSPTK + LPDDFTCGVLAWNSNNLDSK	C48-C101	3059.43	3059.41	3059.42	3059.43	3059.41
LNDLCFTNVYADSFVIR + VVVLSFELLHAPATVCGPK	C60-C194	3966.07	3966.07	3966.07	3966.07	3966.07
DISTEIYQAGSTPCNGVEGFNCYFPL + QSYGFQPTNGVGYQPYR	C149-C157	4765.14	4765.13	4765.14	4765.12	4765.12

**Table 5 vaccines-11-01602-t005:** Comparison of association rate constant (kon), dissociation rate constant (koff), and equilibrium dissociation constant (KD) values of four RBD spy tag batches.

	kon (1/MS)	koff (1/S)	KD (M)
12680T002	4.81 × 10^5^	1.75 × 10^−2^	3.63 × 10^−8^
12680T003	4.15 × 10^5^	2.29 × 10^−2^	5.52 × 10^−8^
12680T004	4.08 × 10^5^	2.35 × 10^−2^	5.77 × 10^−8^
12680T005	2.74 × 10^5^	1.83 × 10^−2^	6.67 × 10^−8^

## Data Availability

The data presented in this study are available on request from the corresponding author.

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
