# Peer review of "Large-Scale Purification and Characterization of Recombinant Receptor-Binding Domain (RBD) of SARS-CoV-2 Spike Protein Expressed in Yeast"

_vaccines, 2023, doi:10.3390/vaccines11101602_

Round 1
Reviewer 1 Report
This manuscript describes a method using yeast expression and SpyTag/Spy Catcher technology to produce high yields of recombinant receptor-Binding domain (RBD) of SARS-CoV-2 to prepare a vaccine against COVID-19. There are several issues with the description of the methodology that require clarification. In addition, there is a lack of positive controls that make it impossible to interpret the significance of many of their findings. My concerns are detailed as follows:
Line 92: The use of “SpyTag/Spy Catcher” technology in this study as described is not clear. A Spy Tag is apparently fused to RBD, however, there is no mention of its use for affinity chromatography (or otherwise) or any mention of a Spy Catcher in the study.
Line 107 https://www.na-107 ture.com/articles/s41598-020-78711-6 refers to a paper published in 2021 and should be included in the list of references rather than a URL. More important, in this publication the yield of RBD from the mammalian expression system was “32 ± 8mg/l” whereas the authors of the paper under review state the mammalian expression system yield is ~5mg/ml compared to their yield of ~20 mg/L using the Pichia pastoris expression system. Even if they confused liters and milliliters and their yield was 20 mg/ml they are still underperforming compared to the mammalian expression system and the main argument in favor of their method.
Line 135: “a peptide composed of 13 amino acids from Streptococcus pyogenes, termed as SPYTAG was fused with gene coding for RBD”. Do the authors mean that a peptide was fused with the RBD gene or that DNA encoding the peptide was used for the construct? They need to identify the corresponding S. pyogenes gene.
Line 138: no reference or source is provided for the methanol-free strain of Komagataella phaffii.
Line 143: The website provided, which I assume refers to doi: 10.1016/j.jiph.2021.06.017, does not specify the RBD sequence and it is unclear why the authors of the manuscript under review selected their exact peptide (range of amino acids).
Line 184: “Operating steps with corresponding buffer and volume details are provided ahead in table 1.” No such table is provided.
Figure 2. The recombinant RBD appears relatively pure and consistent between batches, however, the source of the control, “in-house standard” is not described. The claim of >99% purity, based on the SDS-PAGE (line 191), is questionable without scanning densitometry data and given the reactivity (smearing) in the western blot (Figure 2 B).
Figure 4. The regions of the peptide that correspond to RBD should be specified.
Figures 6, 7 and 8 and Tables 1, 2, and 3. The structure/ expected results of native RBD should be provided for comparison in the results and/or discussion.
The grammar requires extensive editing.
Author Response
Dear Prof./Dr. Reviewer,
Thank you for giving me the opportunity to submit a revised draft of my manuscript titled Large-Scale Purification and Characterization of Recombinant Receptor-Binding Domain (RBD) of SARS-CoV-2 Spike Protein Expressed in Yeast to MDPI journal, Vaccine. We appreciate the time and effort you have dedicated to providing valuable feedback on our manuscript. We are grateful for the efforts of your insightful comments on our paper. We have been able to incorporate changes to reflect most of your suggestions. Your review of the manuscript was immensely helpful. Your suggestions were carefully considered, and as a result, we have thoroughly restructured and updated the manuscript. Thank you for your valuable insights.We have highlighted the changes within the manuscript in BLUE.
Thank you,
Warm Regards

Reviewer 2 Report
A brief summary
This paper aims to report a large-scale production, purification and formulation of SARS-CoV-2 Spike glycoprotein Receptor Binding Domain (RBD) fragment in yeast Pichia pastoris expression system. Expressed and purified RBD protein construct is designed to develop a clinical-stage formulation of the vaccine candidate. The topic is relevant and this study might contribute to the development of technological knowledge. However, a large part of the Results and Methods is missing or insufficiently described. Moreover, the study must be presented in the context of previously published work on the same RBD construct in P. pastoris system. Until this is done, it is difficult to properly assess the scientific value of the work.
General concept comments
The manuscript has serious flaws and cannot be accepted or even properly evaluated in its current form. A very major revision is needed to enable further review.
MAJOR POINTS:
1) References in the manuscript are not correct.
2) Expression vector, expression construct and yeast strain are not described.
3) Important details on fermentation and purification are not provided
4) Control RBD from mammalian cells is required in characterization experiments
5) Methods, Results and Discussion for RBD conjugation with Hepatitis B Surface Antigen virus-like particles followed by adsorption on alum (as stated in Abstract) are missing
6) Supplementary Materials are not accessible at given link.
7) No functional data (e.g. immunization of mice) is provided to base the clinical stage of development of proposed vaccine candidate.
8) Claim on remarkably higher yield of RBD in P. pastoris over mammalian production platforms is not correct.
9) This work seems to be a continuation of previously published studies on the same or very similar RBD constructs in P. pastoris. It should be provided in this context to be understandable.
These points are elaborated in specific comments below.
Specific comments
INTRODUCTION
There is something wrong with citations in this paper. It is not possible to review a background of the topic, because Introduction just cites first nine references (most of them are incorrect), and then no references are given further throughout the manuscript. References include 45 citations in total, but [10-44] of them are not cited in the manuscript at all. Although mentioned unused citations are in the field, but its not clear what statements they support. This should be managed.
In the current manuscript, review of the introduction would be follows:
Lines 59-60: RBD composition in SARS-CoV-2 is described by giving citations to old papers (dated 2005) on SARS-CoV, a different virus (citations [3,5]). Moreover, Figure S1 is not possible to find, as Supplementary Materials was not opened at a given link www.mdpi.com/xxx/s1 (Error 404 - File not found).
Lines 60-63: citations [5-7] are not related to what they cite (regarding long and expensive RBD protein manufacture in mammalian or insect cell lines).
Lines 63-65: “Some of the reports indicate suitability of E.coli, …” – no reports are cited.
Lines 66-67: again, [2] citation is not related to claimed statement regarding RBD expression in E. coli.
Lines 67-69: again, references [4,5] are wrong citations as they do not mention anything about yeast expression systems and alternative vaccine production platforms.
Lines 69-72: no references are given to base yeast as a host to produce recombinant subunit vaccines.
Lines 76-78: reference to [4] is wrong again. Cited paper does not even mention yeast, let alone engineered yeast strains, circulating RBD variants secretion in yeast, etc.
Lines 79-81: references [4,8] do not mentioned neither Pichia pastoris, neither yeast in general.
Lines 81-83: “… we worked on specifically tailored methanol-free yeast stain[4].”. Reference [4] is not related to what is claimed here. Moreover, all authors of cited paper [4] are different than authors of the current submission, hence it is not clear what means the claim “we worked” in this context.
A correct reference [45] is given regarding RBD expression in Pichia and mammalian production platform (lines 107-108), however I’d like to argue that this underestimates RBD expression yields in mammalian expression systems. Although it is possible to find a few initial papers published on RBD expression in mammalian cell lines with reported low secretion yields of 5-10 mg/l, this is misleading. In fact, the RBD secretion yields in HEK293 and CHO expression systems can be much higher. Commercially available Expi293 and ExpiCHO expression systems are very efficient for the secreted RBD monomer production. Here are publications reporting 40-100 mg/L yields of RBD secreted in HEK293 just by transient expression in shake-flasks (further increased in stable fed-batch processes):
García-Cordero J, Mendoza-Ramírez J, Fernández-Benavides D, Roa-Velazquez D, Filisola-Villaseñor J, Martínez-Frías SP, Sanchez-Salguero ES, Miguel-Rodríguez CE, Maravillas Montero JL, Torres-Ruiz JJ, Gómez-Martín D, Argumedo LS, Morales-Ríos E, Alvarado-Orozco JM, Cedillo-Barrón L. Recombinant Protein Expression and Purification of N, S1, and RBD of SARS-CoV-2 from Mammalian Cells and Their Potential Applications. Diagnostics (Basel). 2021 Sep 30;11(10):1808. doi: 10.3390/diagnostics11101808. PMID: 34679506; PMCID: PMC8534734.
Green EA, Hamaker NK, Lee KH. Comparison of vector elements and process conditions in transient and stable suspension HEK293 platforms using SARS-CoV-2 receptor binding domain as a model protein. BMC Biotechnol. 2023 Mar 7;23(1):7. doi: 10.1186/s12896-023-00777-7. PMID: 36882740; PMCID: PMC9990576.
In our laboratory, we have also tried secreted expression of RBD in both mammalian and yeast cells. In mammalian cells, the yields of secreted RBD in our hands were ~180 mg/L using Expi293, and ~430 mg/L using ExpiCHO systems in the shake-flasks (not published). In contrast, secretion in yeast systems was problematic with no secretion in Saccharomyces cerevisiae and with poor results in Pichia. Therefore, the statement in lines 105-107, claiming that this study with ~20 mg/L of RBD from P. pastoris in a bioreactor demonstrate remarkably higher yield compared to mammalian production platforms, is not true and should be avoided.
METHODS
No description is given for the expression vector. It is just mentioned in the Introduction that authors worked “on specifically tailored methanol-free yeast stain”. I was not able to find any accessible publication on this in a list of References. What kind of vector would be for “methanol free” P. pastoris? Is it with HIS4 marker, option to select multicopy strains, what is the promoter (not that of AOX1?)? Why RBD in this study was secreted at all, what was the secretion signal? How authors fused RBD sequence to secretion signal, and how the expression strain was generated (selection of clones, expression check, etc)? Genotype, phenotype, and the source of strain must be clearly described. I was not able to find information on mentioned strain “AltHost S-380” in the internet. Expression construct is not described, but may be inferred from sequence in Fig. 4. It seems that authors expressed RBD sequence comprising 332-532 aa of the Spike protein, which at the C-terminus was fused to glycine-rich 12 aa stretch of GGDGGDGGDGGA (what is this? some linker?) followed by 13 aa of Spytag at the C-terminus. This is unusual construct and rises multiple questions. E.g., RBD sequence is truncated (normally, it comprises aa 319-541 with two N-glycosylation sites at 331 and 343) and authors omitted the first glycosylation site at N331 in their construct. In cited previous studies, both glycosylation sites were expressed, either in P. pastoris or in mammalian cell lines. The question is if such removal of glycosylated sequence affects RBD solubility and structural stability (as authors mentioned in regard of RBD glycosylation in the Introduction section) or not? Next, the vaccine is intended to be used in humans. What would be the role of additional aa sequences included? Is the SpyTag a cause for concern (e.g. immunogenic itself) or not? All rationale for the selected expression construct needs to be provided, as this is essential information for such kind of study.
Methods do not include any citations. In the case of fermentation, specific details are not provided (e.g. the exact composition of BMGY medium, was the culture fed with glycerol or salts, etc). No information what was the production of biomass in such fermentation (OD, or dry cell weight, etc). There is just a scheme for fermentation and purification provided in Fig. 1, which do not contain specific details.
Harvesting of the culture includes strange first step with addition of mild denaturant urea, as authors state “to increase the protein stability”. In our hands, and in previously published studies, addition of urea was not required, neither in mammalian cells nor in P. pastoris culture supernatants (e.g. see cited paper in References [32,45]). This rises concern what was different with expressed specific construct in this study. What would happen if urea was not added (e.g. RBD aggregates, degrades or what exactly happens?)?
No details are given for “hydrophobic interaction chromatography” step which is mentioned in Abstract and showed in Fig. 1. No details on protein recovery and purity in different steps are provided. Taken together, there are so much details missing in various steps that it would not be possible to reproduce the results based on the details given in the methods section.
RESULTS
In principle, the results are limited on characterization of RBD produced in different batches. This is not enough. The main question for the study would be if produced protein maintains native properties attributed to the RBD fragment. It is required to have control recombinant RBD produced by secretion in mammalian cells. E.g., please see cited previous study [32]. It seems that current study used different RBD construct with removed glycosylation site. Therefore, the effect of such approach on RBD properties such as glycosylation, structural stability, folding, and functional performance needs to be explored by comparison to usual “verified” RBD control.
Fig. 2 is not convincing, simply because it shows “too nice” RBD bands. This is in contrast with previous reports. I suspect that Fig. 2 possibly shows RBD fragments after removal of glycans by PNGase treatment. Typically, P. pastoris produces RBD as a broad highly diffused band of glycosylated protein in SDS-PAGE that is much higher on the gel (at ~40-50 kDa) than from mammalian cells (for example, please see Fig. 2 and Fig. 3 in ref. [32]). I suggest to include purified intact and glycosidase-treated RBD samples on the same gel to see the real extent of glycosylation in purified RBD protein.
Fig. 3 just shows molecular masses of deglycosylated RBD preparations. The question is what is the mass of glycosylated RBD, which is finally intended to be used in humans? There should be some mass spectra of purified intact RBD visible by the same ESI-MS approach, considering very nice bands/peaks of the purified glycosylated RBD protein generated by SDS-PAGE and SEC-HPLC (Figs. 2 and 5).
What happened with conjugation of produced RBD to HBsAg VLPs? This data is missing in the Results section. It should be either added or these claims removed from the Abstract and Introduction:
Lines 38-40: “Purified RBD was then conjugated with Hepatitis B Surface Antigen virus-like particles followed by adsorption on alum to develop a clinical-stage formulation of the vaccine candidate.”
Lines 92-93: “In this study, we used a plug-and-display technology called SpyTag/Spy Catcher applied to produce a RBD-based subunit vaccine against COVID-19.”
Finally, Supplementary Materials are not accessible at given link www.mdpi.com/xxx/s1
DISCUSSION
No functional data (e.g. immunization of mice) is provided or discussed to base the clinical stage of development of the proposed vaccine candidate. Still, there is a link (Line 422) to a paper of Dalvie et al., 2021 (a):
Dalvie et el. Scalable, methanol-free manufacturing of the SARS-CoV-2 receptor-binding domain in engineered Komagataella phaffii. Biotechnology and Bioengineering, First published: 15 November 2021. https://doi.org/10.1002/bit.27979
In that paper there is a citation to other previous paper of Dalvie et al., 2021 (b):
Dalvie et al. Engineered SARS-CoV-2 receptor binding domain improves manufacturability in yeast and immunogenicity in mice. Proc Natl Acad Sci U S A. 2021 Sep 21;118(38):e2106845118. https://doi.org/10.1073/pnas.2106845118
In the latter, there is the data on mice immunizations with RBD, etc. Still, it is not clear what is the connection of this study with those previous works that are not cited in the References. It seems that in mentioned study published in PNAS journal authors have used some different RBD construct having mutations that reduce aggregation, improve yield, etc. Here, it seems that RBD construct has a WT sequence without those “improving” mutations (as shown in Fig. 4).
Taken in account all said above, the things should not be presented in such a way. Please, manage your manuscript as required, including the following:
A) Give references to previous relevant works to which this study is connected. Those are mainly published by Dr. Neil C. Dalvie and Prof. J. Christopher Love from Cambridge, Massachusetts, USA. It should be possible to refer to used strain, expression construct, and other specific details if they are the same as in the previous studies.
B) It should be clearly stated how this work is different from previous studies (e.g. from Dalvie et al., 2021a linked in the Discussion) and what new findings it provides.
C) Details that are different and specific to current work need to be clearly described not to be confused with results of previous publications.
D) If previous studies have been carried out with the same RBD construct (initial Wuhan WT sequence of aa 332-532?) in experimental animals, it should be clearly referenced to that specific part of previously published report (it seems there were many different constructs probed including mutated RBDs, conjugated to specific nanoparticles, etc).
E) This paper claims in the Abstract that “formulation of RBD of SARS-CoV-2 protein (…) is currently in the clinical stage of development” (lines 28-29), whereas the Abstract of previous publication of Dalvie et al. (2021a) mentioned that “This engineered strain is now used to produce an RBD-based vaccine antigen that is currently in clinical trials”. Please elaborate which construct in what clinical trials is used, and how the current study contributes to further development.
I see one of the authors of previous studies is included as a co-author in this paper, thus it should not be difficult to give appropriate context of the previous work. Here is good opportunity to manage technical aspects of the previously published studies, because it does not seem easy to find some technical details there. It would help other researchers to benefit from this work and have access to improved strains of P. pastoris in future. This information can improve the significance of the work if provided in appropriate way.
It is not possible to review the manuscript properly until such rewriting is done.
CONCLUSIONS
As mentioned above, statement on the “remarkably higher yield of ~20mg/L in contrast to ~5mg/mL in mammalian production platforms” (Lines 461-462) is not correct and should be changed or removed. Until multiple major points are addressed, closing conclusion (in Lines 466-468) is an overstatement.
Although most of the manuscript is readable, it is worth editing by native English speaker. Besides grammar, there are some irregular sentences that seem strange, for example:
Lines 89-90
Lines 135-141
Lines 143-145
Lines 159-165
Lines 306-307
Lines 311-312
Lines 330-333
Lines 354-355
Lines 367-369
Lines 399-402
Lines 425-429
Lines 435-437
Lines 453-455.
Also, it should be checked for mistakes when giving references to Tables and Figures (e.g. lines 184-185) or for inappropriate use in regard of concentration/volumes (e.g. confusing statement in line 163: “addition of sorbitol to a final Fermenter volume of 1 % “).
Author Response
Dear Prof./Dr. Reviewer,
Thank you for giving me the opportunity to submit a revised draft of my manuscript titled Large-Scale Purification and Characterization of Recombinant Receptor-Binding Domain (RBD) of SARS-CoV-2 Spike Protein Expressed in Yeast to MDPI journal, Vaccine. We appreciate the time and effort you have dedicated to providing valuable feedback on our manuscript. We are grateful for the efforts of your insightful comments on our paper. We have been able to incorporate changes to reflect most of your suggestions. Your review of the manuscript was immensely helpful. Your suggestions were carefully considered, and as a result, we have thoroughly restructured and updated the manuscript. Thank you for your valuable insights. We have highlighted the changes within the manuscript in BLUE.
Thank you,
Warm Regards

Reviewer 3 Report
Reviewer Comments:
- Abstract Clarity and Structure: The abstract effectively highlights the significance of the research and the main findings. However, the abstract could benefit from a clearer structure with distinct sections such as "Background/Objectives," "Methods," "Results," and "Conclusions." This would help readers quickly grasp the study's context, approach, outcomes, and implications.
- Introduction and Context: While the abstract briefly introduces the importance of SARS-CoV-2 Spike protein in COVID-19 vaccine development, it could provide more context on the urgent need for efficient and cost-effective vaccine production methods, especially considering the ongoing global pandemic. Mentioning the limitations and challenges of existing production systems would emphasize the significance of the yeast expression system.
- Methodology Details: It would be beneficial to mention key aspects of the Pichia pastoris expression system, the steps involved in the chromatography-based purification, and the rationale for using mixed mode and hydrophobic interaction chromatography. Additionally, information about the process parameters and conditions used in the large-scale fermentation would enhance the methodological clarity.
- Results Emphasis: The abstract dedicates significant space to describing the results of the study, which is commendable. However, it might be more effective to focus on summarizing the most significant findings in this section and referring readers to the full paper for detailed results. Highlighting the impressive increase in yield from the yeast-based system and the structural and functional characterization could be more impactful.
- Statistical Data and Significance: The abstract mentions the yield of RBD protein and its remarkable batch-to-batch similarity, but specific statistical data (e.g., standard deviations, p-values) would enhance the credibility of these claims. This would provide readers with a clearer understanding of the statistical significance of the results.
- Comparison and Contextualization: While the abstract notes the yield difference between the yeast expression system and mammalian cell systems, it would be valuable to include a brief discussion about the broader implications of this finding. How does this yield improvement impact vaccine production scalability, cost-effectiveness, and potential global vaccine distribution?
- Structural Characterization: The abstract highlights several structural characterization techniques, such as LC-MS peptide mapping, circular dichroism spectroscopy, and fluorescence spectroscopy. While it's important to mention these techniques, it would be helpful to provide a concise summary of the structural insights gained from each method. For instance, any observed conformational changes or structural stability improvements could be highlighted.
- Binding Affinity Insights: The abstract mentions the binding affinity of RBD with ACE2 receptors and confirms consistency among batches. However, a brief discussion about the significance of these results in terms of vaccine efficacy and potential implications for the vaccine's mode of action could enhance the overall understanding.
- Conclusion Emphasis: The conclusion could be more succinctly stated, emphasizing the main takeaway from the study—successfully producing RBD using a yeast-based system with high yield, maintaining structural integrity, and demonstrating consistent binding affinity. This would leave a strong impression on the reader.
- Citations and References: The abstract references some sources for validation and context. Ensure that these references are properly cited in the main text and that any relevant citations from the main body are reflected in the abstract.
Overall, the abstract provides valuable insights into the research, but organizing the content more effectively, emphasizing key findings, providing additional methodological details, and contextualizing the results could enhance its impact and readability.
research manuscript demonstrates generally good English grammar, but there are a few areas where clarity and correctness could be improved by through proof reading prior to re submission
Author Response

(The authors gave the same response as above.)

Round 2
Reviewer 1 Report
The manuscript has been substantially improved.
Minor points:
References should be added to lines 79-82.
Line 105: Higher than what?
Lines 149-152: These two sentences require clarification.
Line 166: Specify wavelength use for OD reading.
Line 368: Specify the source of the in-house standard.
Line 467: Spike not spick protein.
The English grammar needs extensive editing for clarity.
Author Response
Dear Reviewer,
Thank you for allowing me to update and submit a revised draft of my manuscript titled “Large-Scale Purification and Characterization of Recombinant Receptor-Binding Domain (RBD) of SARS-CoV-2 Spike Protein Expressed in Yeast” to MDPI journal, Vaccine. We appreciate the time and effort you have dedicated to providing valuable feedback on our manuscript. We are grateful for the actions of your insightful comments on our paper. We have been able to incorporate changes to reflect most of your suggestions. We have highlighted the changes within the manuscript in BLUE.
Thank you for pointing out the need for correction in English. The manuscript has been reviewed and corrected for English by Professor J. Christopher Love of MIT and Dr. Sergio A. Rodriguez-Aponte (MIT) in the United States. We are open to receiving your suggestions for improving the manuscript if you come across any areas that need improvement.
Your review of the manuscript was immensely helpful. We carefully considered your suggestions, and as a result, we have thoroughly restructured and updated the manuscript. Thank you for your valuable insights.
Thank you,
Warm Regards

Reviewer 2 Report
Thanks for the Authors, manuscript has been improved to enable more appropriate review and evaluation. It is not clear if this paper brings sufficient novelty and value to be published. The weakness of this work is its narrow focus on the production of the antigen at a large scale, which was reported before. Production at 1200 L scale was already mentioned in the previous publication of Dalvie et al., 2021 [19]. There was a brief data from such large-scale purification and the same yield (21 mg/L) reported. Moreover, there are other multiple publications by the authors on the use of this RBD construct. On the other hand, technical side of the expression, purification and analysis was not sufficiently detailed in the previous papers. These details would be worth to publish. However, the current version of the manuscript still misses some important points. There are also some discrepancies or mistakes present in the data. A major revision is required.
MAJOR POINTS:
1) Expression vector, expression construct and yeast strain are not properly described.
2) Some details and results on fermentation and purification are still missing.
3) Differences between batches in Fig. 2A should be explained.
4) Fig. 4 shows the same data for different batches, something is wrong with it.
5) Clinical trials are just mentioned in abstract. It should be elaborated further in manuscript.
6) Discussion on glycosylation of this RBD construct in P. pastoris needs to be included.
MINOR POINTS:
Some references in the manuscript are not correct.
No context is given for claimed higher binding affinity and higher titer of neutralizing antibodies.
Please see specific comments below.
Specific comments:
ABSTRACT
Line 30: “… is currently in clinical trials”. There is no information on this in the manuscript. There should be a registration number and a link (if available) to the trials provided. Authors mentioned in their response that Phase 2 trials in Australia has been successfully completed. It should be stated in the manuscript and trial’s results briefly commented (e.g. how efficiently antibodies were induced by the vaccine, what about side effects, etc.). This elaboration would improve an impact of the work, as manuscript currently does not provide other functional data or comparison to mammalian RBD control.
INTRODUCTION
Although citations have been improved, there are still some minor mistakes and a few incorrect references given.
Line 67: abbreviation LMICs is not required, as it is not used further in manuscript.
Line 70: ref. [13] is not related with what is stated.
Multiple issues with ref. [10]:
Lines 73-75 is about SARS-CoV-2, whereas [10] is about SARS-CoV-1. Moreover, [10] does not mention glycosylation and disulfide bonding.
Lines 77-79: citing of [10] is not correct (there is nothing about yeast).
Lines 459-461 (Discussion): [10] is again not on the topic (would be good for SARS, not for COVID-19).
Lines 86-88: citations are not correct. They suppose that yeast strain expressed mutant RBD variants (alpha, beta, etc), but [16] reports just Wuhan RBD expression, whereas [18] was published before the COVID-19, thus it could not propose anything for this virus.
METHODS
Information on yeast strain and expression construct is inappropriate. Although methods now cite multiple references, they do not stand for this. There is difficult and tiresome to find required information on the strain and vector. This paper is devoted just to the expression, purification, and characterization; therefore, it is necessary to manage these aspects here. No need to describe how improvements of the strain were carried out (previous papers reported on this), but please clearly describe the final producent used in this study. It should include:
A) Strain with fully described genotype. It should be clear what genes were introduced or inactivated in the used strain. Starting with NRRL Y-11430 (was it also modified to start with some (RCR2_D196E, RVB1_K8E), or no?), there should be genotype provided for modifications made on constitutive overexpression of transcription factors mit1 and mxr1, as well as for och1. For the latter, Dalvie et al., 2019 [24] describes its knockout that would be reasoned in this case to deal with hyperglycosylation of RBD in P. pastoris. Here, it is not possible to understand if it was knock-outed or not. No mention if the strain was Mut+ or MutS. Maybe something else (e.g. ku70 knock-out)? Please provide the genotype to answer these questions. Also, note if the strain was selected with multiple copies of the RBD expression construct integrated.
B) Expression vector scheme should be provided. A mention of a “personalized vector” (line 144) is not enough. Ref. [24] reported on other topic, and there were other proteins expressed. This is confusing to seek for the vector of this study in that previous report (and I did not find it).
C) At least, the entire expressed amino acid sequence should be provided, starting from the first Methionine (ATG start of translation). It should include the secretion signal and all expressed sequence till the STOP of translation. It is difficult and troublesome to find the sequence in cited previous studies. There is just a supporting .gb file in ref. [19] where nucleotides for a long alpha-factor secretion signal are indicated. However, it seems that construct did not have sequence for a current linker GGDGGDGGDGG included. There may be issues with correct cleavage of the alpha-factor signal in yeast, etc. Thus, please provide the information on the expressed sequence properly. Ideally, it should indicate all parts: a secretion signal (with pre-, pro- parts, if used), RBD construct, linker, Spy-tag, etc.
Description of fermentation ends with mentioned OD of ≥40 AU before the induction with sorbitol. It seems rather low for bioreactor fermentation of P. pastoris. Please provide what was the OD and cell weight at the harvesting of culture. What was the volume of the supernatant with RBD in a typical run (before or after addition of urea)?
What are the units for Column Volumes (CV) given in Tables 1 and 2 (05 to ~25)?
What were the volumes of purified material before the concentration to final drug substance (lines 212-216)? Table S9 provides final concentration of >10 mg/ml. How much times it needed to be concentrated to this and what is the final volume of a typical drug substance batch?
Abstract (line 33), Introduction (line 105), Discussion (line 470) and Conclusion (line 525) state the 21 mg/L yield of the RBD. This number appears with no data given. It seems just taken from ref. [19] where that yield was already reported. Please provide data how you arrived to this yield in the current study (protein concentration was measured by what means, at which stage, what was the volume of the batch with measured concentration, etc). As this study provides purely technological knowledge, the yield should not pop out from nowhere.
RESULTS
Figures 2A and 3 show different results. In Fig. 2A, batches 2 and 3 migrate slightly faster, whereas batch 4 – considerably faster, than batch 1. Deglycosylated batch 4 still migrate clearly faster than others (lane 9 in Fig. 2A). It most likely contains smaller RBD protein product with remarkable differences of ~4-5 kDa, as authors estimated (lines 346-348)? However, Fig. 3 shows no such differences (despite it was run on the same 12% SDS-PAA gel?) and Fig. 4 shows practically identical MWs among the batches. How that can be? Please explain.
Figure 4: It seems that AB panel is identical to CD panel. In my opinion, it is not possible to get identical data from different samples with this method in such a way. I think that one panel was just copy-pasted from another. And the exact identical MWs given in the picture do not correspond to those given in the Discussion (lines 480-482, MWs of all batches are slightly different, not identical). Please either provide good resolution Fig. 4 for further inspection or correct the figure (e.g., add the true data for batches T002 and T003).
Minor mistake in line 440: Table S2-S5 should Table S5-S8.
DISCUSSION
The essence of this paper requires deeper discussion on the glycosylation of the RBD construct. What is unusual in this work, it shows considerably less extensive glycosylation than other published studies on the RBD expression in P. pastoris. Authors needs to emphasize that they omitted the first glycosylation site in their construct (N331), and therefore final product is overall less glycosylated. It seems a single glycosylation site N343 was enough to maintain required properties of the RBD protein? Another issue is N-glycosylation type in producent strain. Was there a high-mannose glycosylation notable or not? If authors used och1 mutant strain (see, e.g. ref. [24]) with limited glycosylation, it needs to be underscored and commented. It would anyway be in place to discuss if och1 knockout (or other engineering, e.g. ku70 knockout) might be helpful here, or was not required in this approach.
Line 508: “A higher binding affinity …” – higher than what? It should be compared to some expected KD values (measured in other studies?)?
Lines 513-514: “a higher titer of neutralizing antibodies (>104).” – again, higher than what?
CONCLUSIONS
Lines 524-525: “a remarkably higher yield of ~21mg/L.” – remarkably higher over what? Why it can’t be just a “high yield”?
Minor editing of English language required
Author Response
Dear Reviewer,
Thank you for allowing me to update and submit a revised draft of my manuscript titled “Large-Scale Purification and Characterization of Recombinant Receptor-Binding Domain (RBD) of SARS-CoV-2 Spike Protein Expressed in Yeast” to MDPI journal, Vaccine. We appreciate the time and effort you have dedicated to providing valuable feedback on our manuscript. We are grateful for the actions of your insightful comments on our paper. We have been able to incorporate changes to reflect most of your suggestions. We have highlighted the changes within the manuscript in BLUE
The manuscript has been reviewed and corrected for English by Professor J. Christopher Love of MIT and Dr. Sergio A. Rodriguez-Aponte (MIT) in the United States. We are open to hearing your suggestions for improving the manuscript if you come across any areas that you need to improve.
Your review of the manuscript was immensely helpful. We have carefully considered your suggestions, and as a result, we have thoroughly restructured and updated the revised manuscript. Thank you for your valuable insights.
Thank you
Warm Regards

Reviewer 3 Report
I wish to commend the author for addressing all my comments and I now have no further comments.
Please do careful final proof reading
Author Response
Dear Reviewer
Thank you for allowing me to update and submit a revised draft of my manuscript titled “Large-Scale Purification and Characterization of Recombinant Receptor-Binding Domain (RBD) of SARS-CoV-2 Spike Protein Expressed in Yeast” to MDPI journal, Vaccine. We appreciate the time and effort you have dedicated to providing valuable feedback on our manuscript. We are grateful for the actions of your insightful comments on our paper. We have been able to incorporate changes to reflect most of your suggestions.
We appreciate you bringing up the importance of proofreading. The manuscript has been reviewed and corrected for English by Professor J. Christopher Love of MIT and Dr. Sergio A. Rodriguez-Aponte (MIT) in the United States. We are open to hearing your suggestions for improving the manuscript if you come across any areas that you need to improve.
Your review of the manuscript was immensely helpful. We carefully considered your suggestions, and as a result, we have thoroughly restructured and updated the manuscript. Thank you for your valuable insights.
Thank you
Warm Regards
Round 3
Reviewer 2 Report
The manuscript has been considerably improved. Most of the raised points have been addressed, discrepancies or mistakes in the data corrected and managed. Still, I have minor points related to the authors responses that were not included in the revised paper. I believe these sections should be fixed before acceptance of the manuscript for publication.
Specific comments:
1) Information on the expression vector, expression construct and yeast strain are not provided in a clear way, as it should be. In their response authors state that all required information of used strain genotype and vector have been updated in the supplementary data of the revised manuscript. There is no such data except of new Figure S16 with amino acid sequence encoded by the expression vector. Please provide in the Supplementary Data, as mentioned in the response:
- Strain genotype;
- Expression vector scheme (some scheme is provided in the response; it should be included in the Supplementary Data, in good readable state);
- Amino acid sequence (new Fig. S16) should be managed. Now it presents all aa sequences of the expression vector, given as one sequence. This is confusing, as different open reading frames are connected and presented as a single sequence. It should be enough to show just the expression construct of RBD (i.e. a separate single ORF consisting of the signal peptide, RBD protein, linker and Spytag). If other aa sequences of the vector stay (may be removed), they should be shown separated from the expressed ORF containing RBD.
Methods section on the expression (2.2.1) should refer to Supplementary Data where appropriate (e.g. to the vector scheme, the Figure providing amino acid sequence, etc). Also, please fix multiple messy citations in this Method section (“[Error! Bookmark not defined.]”).
2) Authors have responded to the comment for discussion on the glycosylation of the RBD construct (Comment 17, regarding glycosylation sites N331 and N343). I think this information is important and the response should be incorporated into Discussion section of the manuscript. Indeed, presented data suggests that N343 was enough to maintain the required properties of the RBD protein. That explains multiple points, indirectly supports conclusions/findings, and therefore such discussion should not be omitted from the manuscript.
Author Response

(The authors gave the same response as above.)
